# Families of Optimal Transport Kernels for Cell Complexes

## Abstract

Recent advances have discussed cell complexes as ideal learning representations. However, there is a lack of available machine learning methods suitable for learning on CW complexes. In this paper, we derive an explicit expression for the Wasserstein distance between cell complex signal distributions in terms of a Hodge-Laplacian matrix. This leads to a structurally meaningful measure to compare CW complexes and define the optimal transportation map. In order to simultaneously include both feature and structure information, we extend the Fused Gromov-Wasserstein distance to CW complexes. Finally, we introduce novel kernels over the space of probability measures on CW complexes based on the dual formulation of optimal transport.

## 1 Introduction

### 1.1 Cell complexes

A CW complex, or cell complex, is a kind of topological space constructed by inductively gluing cells together by attaching maps. As described by Hatcher (2002), initially we have a collection of zero cells $X^0 = \{e_i^0\}_{i=0}^N$. Then for all $j \in \{1, \dots, n\}$ we take a collection of $j$-cells $\{e_i^j\}_{i=0}^N$ and glue their boundaries to points in the $j-1$ skeleton using continuous attaching maps $\phi_i^j : \partial e_i^j \to X^{j-1}$. Essentially, a CW complex is constructed by taking a union of a sequence of topological spaces $\emptyset = X^{-1} \subset X^0 \subset X^1 \subset \cdots$ such that each $j$-skeleton is obtained by attaching $j$-cells to the $(j-1)$-skeleton (Hatcher, 2002).

### 1.2 Optimal transport

Classically optimal transport provides a framework for comparing probability measures. Let $\mathcal{P}(\Omega)$ be the set of all probability measures on a space $\Omega$. Let $\mathcal{B}(\mathcal{X})$ be the set of all Borel sets of a $\sigma$-algebra $\mathcal{X}$. Let $\#$ denote the pushforward such that for $A \in \mathcal{B}(\mathcal{X})$ we have $T_\# \mu(A) = \mu(T^{-1}(A))$.

We provide the following important definitions related to the formulation of optimal transport from Genevay et al. (2016); Vayer et al. (2018); Villani (2021).

**Definition 1.1.** Let $\mathcal{X}$ and $\mathcal{Y}$ be separable metric spaces. Then for measures $\mu \in \mathcal{P}(\mathcal{X})$ and $\nu \in \mathcal{P}(\mathcal{Y})$ define the set of all couplings of $\mu$ and $\nu$ as: $\Pi(\mu, \nu) := \{\pi \in \mathcal{P}(\mathcal{X} \times \mathcal{Y}) \mid \forall (A, B) \in \mathcal{B}(\mathcal{X}) \times \mathcal{B}(\mathcal{Y}), \pi(A \times \mathcal{Y}) = \mu(A), \pi(\mathcal{X} \times B) = \nu(B)\}$ (Genevay et al., 2016; Vayer et al., 2018).

**Definition 1.2.** Let $\mathcal{X}$ and $\mathcal{Y}$ be Polish spaces. Let $c : \mathcal{X} \times \mathcal{Y} \to [0, \infty]$ be a Borel-measurable cost function such that $c$ measures the cost of transporting one unit of mass of $\mathcal{X}$ to $\mathcal{Y}$. Then given $\mu \in \mathcal{P}(\mathcal{X})$ and $\nu \in \mathcal{P}(\mathcal{Y})$ the Monge formulation of optimal transport is to find a $\mu$-measurable map $T : \mathcal{X} \to \mathcal{Y}$ such that $\nu = T_\# \mu$ and we

$$\text{minimize } \mathcal{I}(T) = \int_{\mathcal{X}} c(x, T(x)) d\mu(x) \tag{1}$$

over the set of all $T$ (Vayer et al., 2018; Villani, 2021).

**Definition 1.3.** Let $\mathcal{X}$ and $\mathcal{Y}$ be Polish spaces. Let $c : \mathcal{X} \times \mathcal{Y} \to [0, \infty]$ be a Borel-measurable cost function such that $c$ measures the cost of transporting one unit of mass of $\mathcal{X}$ to $\mathcal{Y}$. Then given $\mu \in \mathcal{P}(\mathcal{X})$ and $\nu \in \mathcal{P}(\mathcal{Y})$ the Kantorovich formulation of optimal transport is to

$$\text{minimize } \mathcal{I}(\pi) = \int_{\mathcal{X} \times \mathcal{Y}} c(x, y) d\pi(x, y) \text{ for } \pi \in \Pi(\mu, \nu). \tag{2}$$

The optimal transportation cost is then $T_{opt}(\mu, \nu) = \inf_{\pi \in \Pi(\mu, \nu)} \mathcal{I}(\pi)$ (Villani, 2021).

**Definition 1.4.** Let $\mathcal{X}$ be a polish space endowed with distance $d$ and let $p \in \mathbb{R}_{\geq 0}$. Let $c(x_i, x_j) = d(x_i, x_j)^p$ with the convention that $d(x_i, x_j)^0 = \mathbb{1}_{x_i \neq x_j}$. Let $\mathcal{T}_p(\mu, \nu) = \mathcal{T}_{d^p}(\mu, \nu)$ for the associated optimal transport cost between Borel measures $\mu, \nu$ on $\mathcal{X}$. We define $\mathcal{P}_p(\mathcal{X})$ to be the set of probability measures with finite moments of order $p$ in essence measures $\mu$ such that $\forall x_i \in \mathcal{X}, \int d(x_i, x_j)^p d\mu(x) < \infty$. If $d$ is bounded then $\mathcal{P}_p(\mathcal{X})$ coincides with $\mathcal{P}(\mathcal{X})$ (Villani, 2021).

**Definition 1.5.** Let $\mathcal{X}$ be a polish space endowed with distance $d$. Then for $p \in [1, \infty)$ the Wasserstein $p$-distance between measures $\mu$ and $\nu$ on $\mathcal{X}$ with finite $p$-moments is $W_p = \mathcal{T}_p^{1/p}$ which defines a metric on $\mathcal{P}_p(\mathcal{X})$. Essentially we can write:

$$W_p(\mu, \nu) = \inf_{\pi \in \Pi(\mu, \nu)} \left( \int_{\mathcal{X} \times \mathcal{X}} d(x, y)^p d\pi(x, y) \right)^{\frac{1}{p}} \tag{3}$$

where $W_p(\mu, \nu)$ is the Wasserstein $p$-distance (Vayer et al., 2018; Villani, 2021).

## 2 Related Work

### 2.1 Gaussian Processes on Topological Spaces

Alain et al. (2023) define a Gaussian process over cell complexes by introducing a generalization of the random-walk graph Matern kernel and a generalization of the graph diffusion kernel, which they term the reaction-diffusion kernel. Yang et al. (2024) define a Gaussian process designed to model functions defined over the edge set of a simplicial 2-complex. The authors explicitly define classes of divergence-free and curl-free edge Gaussian processes. They then combine them with the Hodge decomposition to create Hodge-compositional edge Gaussian processes, which can represent any edge function (Yang et al., 2024). Kerimov et al. (2024) apply edge Gaussian processes to perform state estimation in water distribution systems. Alain et al. (2025) develop a Gaussian process for graph classification that leverages the Hodge decomposition and can leverage vertex and edge features. Alain et al. (2025) build on the work of Opolka et al. (2023) who solve the same problem using spectral features for graph learning. Batko et al. (2024) apply cellular Gaussian processes, specifically the Matern kernel, to characterize behavior in nonlinear dynamical systems. Perez et al. (2024) developed the sliced Wasserstein Weisfeiler-Lehman graph kernels and a corresponding Gaussian process. Borovitskiy et al. (2021) developed the Matern kernel for graph Gaussian processes. CW complexes are generalizations of graphs and simplicial complexes. Optimal transport kernels enjoy numerous theoretical advantages and are computationally efficient (Bachoc et al., 2023; Thi Thien Trang et al., 2021; Pereira & Amini, 2025). We develop kernels over CW complexes that rely on regularized optimal transport and use them to define a Gaussian process. It should be noted that kernels relying on optimal transport are distinct from the already existing random-walk, Matern, and diffusion kernels for cell complexes. Given that we develop optimal transport (OT) based kernels, all the theoretical advantages of OT kernels apply.

### 2.2 Graph Optimal Transport

Petric Maretic et al. (2019) develop a framework for Optimal transport on Graphs. The authors interpret graphs as elements that drive the probability distributions of signals (Petric Maretic et al., 2019). Additionally, Vayer et al. (2019) develop a Fused Gromov-Wasserstein distance for graphs. The Fused Gromov-Wasserstein distance enables determining Optimal Transport distances which include both features and structures (Vayer et al., 2018; 2019). In this section we provide both of their key definitions.

Petric Maretic et al. (2019) provide the following setup defining optimal transport on graphs.

**Definition 2.1.** Suppose we have two graphs $\mathcal{G}_1$ and $\mathcal{G}_2$ with Laplacian matrices $L_1$ and $L_2$ respectively. Additionally, let $L_1^\dagger$ and $L_2^\dagger$ be the Moore-Penrose pseudoinverses of $L_1$ and $L_2$. Moreover, let $\nu^{\mathcal{G}_1} = \mathcal{N}(0, L_1^\dagger)$ and $\mu^{\mathcal{G}_2} = \mathcal{N}(0, L_2^\dagger)$ be the respective normal distributions corresponding to $\mathcal{G}_1$ and $\mathcal{G}_2$. Then the Wasserstein-2 distance with respect to the standard Euclidean norm is

$$W_2(\nu^{\mathcal{G}_1}, \mu^{\mathcal{G}_2}) = \inf_{T_\# \nu^{\mathcal{G}_1} = \mu^{\mathcal{G}_2}} \int_{\mathcal{X}} \|x - T(x)\|^2 d\nu^{\mathcal{G}_1}(x) \tag{4}$$

where $T_{\#}\nu^{\mathcal{G}_1}$ is the pushforward by transport map $T : \mathcal{X} \to \mathcal{X}$ defined on Polish space $\mathcal{X}$. We can additionally write the Wasserstein-2 distance as:

$$W_2(\nu^{\mathcal{G}_1}, \mu^{\mathcal{G}_2}) = \text{Tr}(L_1^\dagger + L_2^\dagger) - 2\,\text{Tr}\left(\sqrt{L_1^{\frac{\dagger}{2}} L_2^\dagger L_1^{\frac{\dagger}{2}}}\right) \tag{5}$$

with optimal transport map $T(x) = L_1^{\frac{\dagger}{2}}\left(L_1^{\frac{\dagger}{2}} L_2^\dagger L_1^{\frac{\dagger}{2}}\right)^{\frac{\dagger}{2}} L_1^{\frac{\dagger}{2}} x$.

Vayer et al. (2019) then develops a Fused Gromov-Wasserstein distance for graphs $\mathcal{G}_1$ and $\mathcal{G}_2$ and a notion of optimal transport on graphs (Vayer et al., 2018; 2019).

**Definition 2.2.** Given $\mathcal{G}_1$ and $\mathcal{G}_2$ and probability measures $\mu = \sum_{i=1}^n h_i \delta_{(x_i, a_i)}$ and $\nu = \sum_{j=1}^m g_j \delta_{(y_i, b_i)}$ where $h \in \sigma_n$ and $g \in \sigma_m$ are histograms. Then with an admissible set of couplings

$$C(h, g) = \left\{ \pi \in \mathbb{R}_+^{n \times m} \,\middle|\, \sum_{i=1}^n \pi_{ij} = h_j \wedge \sum_{j=1}^m \pi_{ij} = g_i \right\} \tag{6}$$

we define the Fused Gromov-Wasserstein (FGW) as:

$$FGW_{p,\alpha}(\mu, \nu) = \min_{\pi \in C(h,g)} \mathbb{E}_p[M_{AB}, C_1, C_2, \pi] \tag{7}$$

where $\alpha \in [0, 1]$, $C_1$ and $C_2$ are the structure matrices for graphs $\mathcal{G}_1$ and $\mathcal{G}_2$ respectively, $M_{AB} = (d(a_i, b_j))_{ij}$ is an $n \times m$ matrix of distances between features, and $L_{ijkl} = |C_1(i, k) - C_2(j, l)|$ is a 4-tensor measuring similarity between all pairwise distances in the graph. We expand out the expectation as follows,

$$\mathbb{E}_p[M_{AB}, C_1, C_2, \pi] = \langle (1 - \alpha) M_{AB}^p + \alpha L(C_1, C_2)^p \otimes \pi, \pi \rangle$$
$$= \sum_{i,j,k,l} (1 - \alpha) d(a_i, b_j)^p + \alpha |C_1(i, k) - C_2(j, l)|^p \pi_{ij} \pi_{kl} \tag{8}$$

thereby defining the FGW distance. Moreover as $\alpha \to 0$ the authors show the FGW distance recovers the Wasserstein distance between features

$$\lim_{\alpha \to 0} FGW_{p,\alpha}(\mu, \nu) = W_p(\mu_A, \nu_B)^p = \min_{\pi \in \Pi(h,g)} \langle \pi, M_{AB}^p \rangle \tag{9}$$

Similarly as $\alpha \to 1$ the FGW distance recovers the Gromov-Wasserstein distance between structures

$$\lim_{\alpha \to 1} FGW_{p,\alpha}(\mu, \nu) = GW_p(\mu_A, \nu_B)^p = \min_{\pi \in \Pi(h,g)} \langle L(C_1, C_2)^p \otimes \pi, \pi \rangle \tag{10}$$

## 3 Optimal Transport for CW complexes

### 3.1 Embedding CW complexes as Polish spaces

In order to ensure that the existing formulations of optimal transport, Wasserstein distance, and Fused Gromov-Wasserstein distance are compatible with CW complexes we describe how to embed a CW complex as a Polish space. Essentially, we want to view finite CW complexes of topological dimension $n$ as probability distributions embedded in some Euclidean space.

**Theorem 3.1.** *If $X$ is a finite CW complex of topological dimension $n$, then we can embed $X$ in a polish space $Y$.*

*Proof.* Suppose we let $X$ be a finite CW complex of dimension $n$. Then by Hatcher (2002) Theorem 2C.5, $X$ is homotopy equivalent to a finite dimensional locally finite simplicial complex $S$ (Hatcher, 2002). Then by Lazarus (2020) we know $S$ has linear embedding $X \in \mathbb{R}^k$ (Lazarus, 2020). Pick regular neighborhood $M_x$

of $X \in \mathbb{R}^k$. It will be homotopy equivalent to $X$ and a complex PL submanifold with boundary. Take the interior $\text{int}(M_x) = \mathcal{M}$. Then $\mathcal{M}$ is a connected smooth manifold approximation of $X$. Then by Whitney's Theorem $\mathcal{M}$ can be properly embedded in a Euclidean space $Y$ (Lee, 2012). In other words we can view the embedding of $\mathcal{M}$ as a closed subset of an Euclidean space with the induced topology. Therefore the embedding of $\mathcal{M}$, namely $Y$, is a Polish space (Whitney, 2012). □

*Remark* 3.2. Alternatively, one can directly embed $X$ in $\mathbb{R}^{2n+1}$ by Menger-Nöbeling theorem (Gilbert, 1972). We can view this embedding as a closed subset of a Euclidean space with the induced topology and therefore a Polish space.

## 3.2 Kantorovich Formulation of Optimal Transport on CW complexes

Using Theorem 3.1, we can write out the Kantorovich formulation of optimal transport which closely follows Definition 1.3.

**Definition 3.3.** Let $X$ and $Y$ be a finite CW complexes of topological dimension $n$. Then embed $X$ and $Y$ as polish spaces using Theorem 3.1, and denote the embeddings $\mathcal{X}$ and $\mathcal{Y}$ respectively. Let $c : \mathcal{X} \times \mathcal{Y} \to [0, \infty]$ be a Borel-measurable cost function such that $c$ measures the cost of transporting one unit of mass of $\mathcal{X}$ to $\mathcal{Y}$. Then given $\mu \in \mathcal{P}(\mathcal{X})$ and $\nu \in \mathcal{P}(\mathcal{Y})$ we can write the traditional Kantorovich formulation of optimal transport:

$$\text{minimize } \mathcal{I}(\pi) = \int_{\mathcal{X} \times \mathcal{Y}} c(x, y) d\pi(x, y) \text{ for } \pi \in \Pi(\mu, \nu). \tag{11}$$

The optimal transportation cost is then $T_{opt}(\mu, \nu) = \inf_{\pi \in \Pi(\mu, \nu)} \mathcal{I}(\pi)$.

## 3.3 Wasserstein Distance for CW complexes

In order to compare CW complexes with varying structures we want to develop a notion of similarity. We develop a notion of transportation distance to compare two CW complexes represented as probability distributions. In order to achieve this we define the Wasserstein distance for CW complexes.

**Definition 3.4.** Suppose we have two finite $n$ dimensional CW complexes $X_1$ and $X_2$. Let $\Delta_{X_1}$ and $\Delta_{X_2}$ be the Hodge Laplacian matrices corresponding to $X_1$ and $X_2$ respectively (Definition A.6). Additionally, let $\Delta_{X_1}^\dagger$ and $\Delta_{X_2}^\dagger$ be the Moore-Penrose pseudoinverses of $\Delta_{X_1}$ and $\Delta_{X_2}$. Let $\mathcal{X}_1$ and $\mathcal{X}_2$ correspond to the polish space embeddings of $X_1$ and $X_2$ respectively (Theorem 3.1). Then by Kechris (2012) we know that the countable product of polish spaces is a polish space (Kechris, 2012). Let $\mathcal{X}$ be the countable product of $\mathcal{X}_1$ and $\mathcal{X}_2$, in essence $\mathcal{X} := \prod_{i=1}^{2} \mathcal{X}_i$. Therefore $\mathcal{X}$ is a polish space. Let $d$ be a valid distance on $\mathcal{X}$. Then we can let $\mu^{X_1} = \mathcal{N}(0, \Delta_{X_1}^\dagger)$ and $\nu^{X_2} = \mathcal{N}(0, \Delta_{X_2}^\dagger)$ be the respective normal distributions corresponding to $X_1$ and $X_2$. Therefore, $\mu^{X_1} \in \mathcal{P}_p(\mathcal{X})$ and $\nu^{X_2} \in \mathcal{P}_p(\mathcal{X})$. As a result, we can define the Wasserstein-$p$ distance as:

$$W_p(\mu^{X_1}, \nu^{X_2}) = \inf_{\pi \in \Pi(\mu^{X_1}, \nu^{X_2})} \left( \int_{\mathcal{X} \times \mathcal{X}} d(x, y)^p d\pi(x, y) \right)^{\frac{1}{p}} \tag{12}$$

When $p = 2$, $d$ is the euclidean norm, $T : \mathcal{X} \to \mathcal{X}$ is a transport map, and $T_\# \mu^{X_1}$ is the pushforward by transport map $T$, we can write the Wasserstein-2 distance:

$$W_2(\mu^{X_1}, \nu^{X_2}) = \inf_{T_\# \mu^{X_1} = \nu^{X_2}} \int_{\mathcal{X}} \|x - T(x)\|^2 d\mu^{X_1}(x) \tag{13}$$

$$W_2(\mu^{X_1}, \nu^{X_2}) = \text{Tr}(\Delta_{X_1}^\dagger + \Delta_{X_2}^\dagger) - 2\,\text{Tr}\left( \sqrt{\Delta_{X_1}^{\frac{\dagger}{2}} \Delta_{X_2}^\dagger \Delta_{X_1}^{\frac{\dagger}{2}}} \right) \tag{14}$$

with optimal transport map $T(x) = \Delta_{X_1}^{\frac{\dagger}{2}} \left( \Delta_{X_1}^{\frac{\dagger}{2}} \Delta_{X_2}^\dagger \Delta_{X_1}^{\frac{\dagger}{2}} \right)^{\frac{\dagger}{2}} \Delta_{X_1}^{\frac{\dagger}{2}} x$.

Instead of comparing CW complexes directly we can look at the signal distributions, as done by Petric Maretic et al. (2019). Specifically we are measuring the dissimilarity between two CW complexes through the Wasserstein distance of their respective distributions.

### 3.4 Fused Gromov-Wasserstein Distance for CW complexes

Optimal transport enables comparison of empirical distributions. In order to additionally compare structural information of we extend the work of Vayer et al. (2019) by defining a fused Gromov-Wasserstein distance for CW complexes.

Alain et al. (2023) provide a framework with which to enrich cell complexes. Namely one can assign cell weights and capture important structural information using boundary matrices (definition A.6). Formally the Hodge Laplacian has matrix representation:

$$\Delta_k := B_k^\top W_{k-1}^{-1} B_k W_k + W_k^{-1} B_{k+1} W_{k+1} B_{k+1}^\top \tag{15}$$

where the $W_i$ are diagonal matrices of cell weights and the $B_i$ are boundary matrices.

**Definition 3.5.** Suppose we have two finite $n$ dimensional CW complexes $X_1$ and $X_2$. Let $\Delta_{X_1}$ and $\Delta_{X_2}$ be the Hodge Laplacian matrices corresponding to $X_1$ and $X_2$ respectively (Definition A.6). Additionally, let $\Delta_{X_1}^\dagger$ and $\Delta_{X_2}^\dagger$ be the Moore-Penrose pseudoinverses of $\Delta_{X_1}$ and $\Delta_{X_2}$. Let $\mathcal{X}_1$ and $\mathcal{X}_2$ correspond to the polish space embeddings of $X_1$ and $X_2$ respectively (Theorem 3.1). Then by Kechris (2012) we know that the countable product of polish spaces is a polish space (Kechris, 2012). Let $\mathcal{X}$ be the countable product of $\mathcal{X}_1$ and $\mathcal{X}_2$, in essence $\mathcal{X} := \prod_{i=1}^2 \mathcal{X}_i$. Therefore $\mathcal{X}$ is a polish space. Let $d$ be a valid distance. Then we can define $M_{k,X_1,X_2} = (d(w_{1,i}^k, w_{2,j}^k))_{ij}$ to be an $k \times k$ matrix of distances between cell weights from the $k$-chain in CW complex $X_1$ and the $k$-chain in CW complex $X_2$ respectively. Then we can let $\mu = \mathcal{N}(0, \Delta_{X_1}^\dagger)$ and $\nu = \mathcal{N}(0, \Delta_{X_2}^\dagger)$ be the respective normal distributions corresponding to $X_1$ and $X_2$. Therefore, $\mu \in \mathcal{P}_p(\mathcal{X})$ and $\nu \in \mathcal{P}_p(\mathcal{X})$. As a result, we can define a kind of Fused Gromov-Wasserstein distance (CW FGW) as:

$$FGW_{p,\alpha}(\mu,\nu) = \min_{\pi \in \Pi(\mu,\mu)} \mathbb{E}_p[M_{k,X_1,X_2}, \Delta_{X_1}\Delta_{X_2}, \pi] \tag{16}$$

where $\alpha \in [0,1]$, and $L_{i,j,k,l} = |\Delta_{X_1}(i,k) - \Delta_{X_2}(j,l)|$ is a tensor measuring similarity between all pairwise distances in the CW complex. We expand out the expectation as follows,

$$\mathbb{E}_p[M_{AB}, C_1, C_2, \pi] = \langle (1-\alpha)M_{k,X_1,X_2}^p + \alpha L(\Delta_{X_1}, \Delta_{X_2})^p \otimes \pi, \pi \rangle$$
$$= \sum_{i,j,k,l} (1-\alpha)d(w_i^k, w_j^k)^p + \alpha|\Delta_{X_1}(i,k) - \Delta_{X_2}(j,l)|^p \pi_{ij}\pi_{kl} \tag{17}$$

thereby defining the CW-FGW distance.

## 4 OT-Kernels for CW complexes

Using the Wasserstein distance and Fused Wasserstein distance adapted to CW complexes we can define a new families of kernels.

### 4.1 CW-Wasserstein Distance and Exponential Kernels

Suppose we have CW complexes $X_1$ and $X_2$, measures $\mu = \mathcal{N}(0, \Delta_{X_1}^\dagger)$, and $\nu = \mathcal{N}(0, \Delta_{X_2}^\dagger)$, and Wasserstein-p distance $W_p$ as in definition 3.4. Then we can define a new family of kernel functions for $p > 1$:

$$k_W = \exp\left(-\frac{W_p(\mu,\nu)}{2\sigma^2}\right) \tag{18}$$

However, this has undesirable consequences for positive definiteness. The $W_p$ as defined in 3.4 is not guaranteed to be conditionally negative definite. As a result we can't guarantee that any resulting squared exponential kernel is positive definite. We overcome this by applying the technique in 4.3.

### 4.2 CW-FGW Distance and Exponential Kernels

Suppose we have CW complexes $X_1$ and $X_2$, measures $\mu = \mathcal{N}(0, \Delta_{X_1}^\dagger)$, and $\nu = \mathcal{N}(0, \Delta_{X_2}^\dagger)$, and Fused Gromov Wasserstein distance $FGW$ as in definition 3.5. Then we can define a new family of kernel functions:

$$k_{FGW} = \exp\left(-\frac{FGW(\mu,\nu)}{2\sigma^2}\right) \tag{19}$$

Just as is the case with the CW-Wasserstein distance, this has undesirable consequences for positive definiteness. The $FGW$ distance as defined in 3.5 is not guaranteed to be conditionally negative definite. As a result we can't guarantee that any resulting squared exponential kernel is positive definite. We overcome this by applying the technique in 4.3.

### 4.3 Enforcing positive definite kernels

Fortunately, applying a technique developed by De Plaen et al. (2020) enables us to choose a bandwidth parameter $\sigma > 0$ such that the resulting Gram matrix of $k_W$ is positive definite (De Plaen et al., 2020). Suppose we have a dataset $\mathcal{D} := \{(x_i, y_i)\}_{i=1}^n$ where $x_i$ is a CW-complex and $y_i \in \mathbb{R}^k$. Let the set $X = \{x_i\}_{i=1}^N$ contain all $x_i$ in our dataset. Additionally, let $d : X \times X \to \mathbb{R}_{\geq 0}$ be a symmetric distance function such that $d(x,x) = 0$. Then we can write out the gram matrix $K_{d,\sigma}$ as:

$$K_{d,\sigma} = \left[ \exp\left( \frac{-d^2(x_i, x_j)}{2\sigma^2} \right) \right]_{i,j=1}^N \tag{20}$$

This matrix is symmetric with real eigenvalues. The minimum eigenvalue $\lambda_{min}(\sigma)$ of $K_{d,\sigma}$ is a function $\lambda_{min} : \mathbb{R}_{>0} \to \mathbb{R}$, $\sigma \mapsto \min\{\lambda_1, \ldots \lambda_N\}$ where $\lambda_i$ are the eigenvalues of $K_{d,\sigma}$.

Then by De Plaen et al. (2020) Proposition 2.4 we know there exists $\sigma_{PSD} \in \mathbb{R}_+$ such that $K_{d,\sigma}$ is positive semi-definite for all $\sigma \leq \sigma_{PSD}$. As a consequence, we can find a finite positive definite approximation for our Wasserstein distance.

Essentially, we can find a finite dimensional feature map $\phi(x)$ such that the positive semi-definite kernel $\phi(x)^\top \phi(x)$ approximates either $k_W$ or $k_{FGW}$ as given in equations (18) and (19) respectively.

The approximation method is given by De Plaen et al. (2020) is specified below in definition 4.1.

**Definition 4.1.** Consider finite dimensional feature map $\phi$ such that $k(x,y) \approx \phi(x)^\top \phi(y)$ (18). This finite approximation is determined by constructing a kernel matrix $K = [k(x_i, x_j)]_{i,j=1}^N$ for set $X$ as in (19). We can truncate the spectral decomposition of $K = \sum_{k=1}^N \lambda_k v_k v_k^\top$ to the $\ell$ largest strictly positive eigenvalues. This results in a new positive definite matrix $K^\ell := \sum_{k=1}^\ell \lambda_k v_k v_k^\top \succ 0$ with $\lambda_1 \geq \cdots \geq \lambda_N$. Therefore we can reconstruct the different components of an approximate feature map:

$$\phi_\ell := \frac{1}{\sqrt{\lambda_\ell}} k_x^\top v_\ell \quad \text{for } i = 1, \ldots, \ell \tag{21}$$

with $k_x := [k(x, x_1) \cdots k(x, x_N)]^\top$. These components are termed Wasserstein features and they compose the approximate feature map $\phi(x) = [\phi_1(x) \cdots \phi_\ell(x)]^\top$ of the kernel in equation 18 (De Plaen et al., 2020).

We apply the method outlined in definition 4.1 to the kernel functions given in (18) and (19). Thereby enabling us to define a Gaussian process over CW-complexes that relies on regularized optimal transport.

## 5 Experiment

In this section we discuss how to determine the optimal transportation cost between CW-complexes. Using our distances, described in equations (12) and (16), we can compare CW-complexes that have the same topological dimension through their signal distributions.

In our approach we are given two CW-Complexes $X_1$ and $X_2$, each of dimension $N$. We aim to find the optimal transportation map $T$ from $X_1$ to $X_2$. However, the topology of each complex need not be the same. We only need to ensure the dimension of the Hodge-Laplacian matrices are equivalent. As above our probability measures are $\mu = \mathcal{N}(0, \Delta_{X_1}^\dagger)$ and $\nu = \mathcal{N}(0, \Delta_{X_2}^\dagger)$. In short we wish to solve the optimization problem:

$$\underset{\theta \in \mathbb{R}^{n \times k}}{\text{minimize}} \, W_2(\mu, \nu) \quad \text{s.t.} \begin{cases} \theta \in \mathbb{R}^{n \times k} \\ P(\nu_i \mid f_\theta(\mu_i), \mu_i) \sim N(\nu_i \mid f_\theta(\mu_i), \sigma^2) \end{cases} \tag{22}$$

In essence, we wish to minimize the Wasserstein-2 distance between our measures $\mu$ and $\nu$. We approach this by defining a map $f$ with parameters $\theta$ which we denote $f_\theta$. $f_\theta$ transports the mass from $X_1$ to $X_2$. Therefore

finding the optimal transportation cost can be approximated by minimizing the exact marginal log-likelihood between $f_\theta(\mu_i)$ and $\nu_i$ for discrete samples. We let $f_\theta \sim \mathcal{GP}(m(x), k(x, x'))$ be our Gaussian Process with mean function $m$ and our kernel function $k$ may either be $k_W$ or $k_{FGW}$ as defined in equations (18) and (19). Essentially, we approximately determine the optimal transport map $f_\theta$ by fitting a Gaussian Process which is well defined over CW-complexes.

---

**Algorithm 1** Approximate solution to the optimal transport problem as defined in (3.3).

---

  **Require:** CW-complexes $X_1$ and $X_2$.
  **Require:** Number of epochs $E$, sampling size $S \in \mathbb{N}$, learning rate $\gamma$, and $p > 1$.
  **Choose:** Kernel function $k$ from $\{k_W, k_{FGW}\}$ and let $m$ be the constant mean function and $K$ be the kernel matrix.
  **Define:** Gaussian process $f_\theta \sim \mathcal{GP}(m, k)$, parameters $\theta \in \mathbb{R}^{n \times k}$ randomly initialized.
  **Construct:** Distributions $\mu = \mathcal{N}(0, \Delta_{X_1}^\dagger)$ and $\nu = \mathcal{N}(0, \Delta_{X_2}^\dagger)$
  **for** $i = 0, 1, \ldots, E$ **do**
    Draw samples $\{\mu_i\}_{i=1}^S$ from $\mu$ and $\{\nu_i\}_{i=1}^S$ from $\nu$.
    Define the approximate cost function as
    $J(\theta) = \frac{1}{S} \sum_{i=1}^S \log(P(\nu_i \mid f_\theta(\mu_i), \theta) P(f_\theta(\mu_i) \mid \mu_i))$
    $\theta^{i+1} \leftarrow \theta^i - \gamma \nabla J(\theta)$
  **end for**
  **return** $\theta$, $f_\theta$, $K$

---

Monge, in 1781, initially motivated optimal transport with a practical scenario about moving matter (Markowich, 2007). Essentially, we are given a pile of stone and must use it to build a castle. We want to find the optimal way to achieve this goal. In essence, we want to transport the pile of stones to the castle in a way that minimizes the total amount of work we must do. In this same vein, one can view our experiment as solving the problem, "When given a pair of cell complexes $X_1$ and $X_2$, what is the most cost-effective way to transform $X_1$ into $X_2$?". In a sense, $f_\theta^*$ is our approximate answer to this question. This is visually depicted below in figure 1.

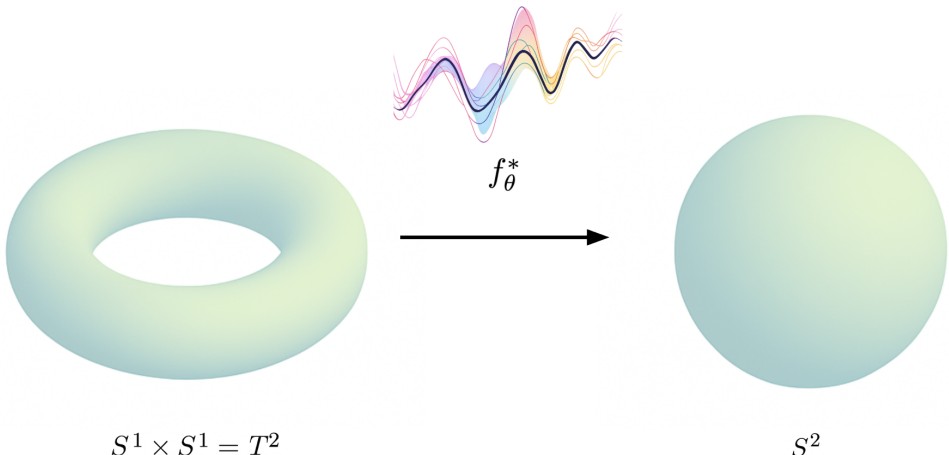

$$f_\theta^*$$

$$S^1 \times S^1 = T^2 \qquad\qquad S^2$$

Figure 1: In the figure above we can see the torus $T^2$ being "transported" via $f_\theta$ into the sphere $S^2$. This is a visual rendition of the problem.

## 5.1 Experimental Setup

We construct a randomized dataset consisting of CW complexes suited for each kernel. In the case of the Exponential Wasserstein Kernel, as defined in (18), we construct a dataset $\mathcal{D} := \{(x_i, y_i)\}_{i=1}^N$ where $x_i$ and $y_i$

are CW-complexes. We additionally include cell features in our dataset for the Fused Gromov-Wasserstein kernel. In essence, $\mathcal{D} := \{(x_i, w_i, y_i)\}_{i=1}^N$ where $x_i$ and $y_i$ are CW complexes and $w_i$ are cell weights (features) corresponding to $x_i$. These features are randomly sampled from a Gaussian. We then fit the Gaussian process using Algorithm 1 and evaluate the model on a holdout set of datapoints. We pick $N = 1000$ and use a 70/30 train/test split. We train for 15 epochs. We visualize the training loss, test loss, and noise parameter per epoch in figure 2. The kernel matrices is visualized in figure 3. As our method is the first to explicitly define optimal transport-based measures directly between CW complexes, direct empirical comparisons with existing methods are currently not possible. Existing approaches, such as the Matern kernels for CW complexes, or graph-based kernels, do not directly generalize to this particular problem.

## 5.2 Experimental Results

In Table 1, we observe that the test set loss of the Fused Gromov-Wasserstein (FGW) kernel is substantially better when compared to the exponential kernel. This indicates that the model is able to leverage the additional features and better model the underlying structure. We observe the training evolution in figure 2. Both methods clearly exhibit stable convergence without overfitting significantly. Notably, the noise parameter increases steadily during training, reflecting a learned balance between structure fidelity and flexibility. From figure 3, we see that the exponential kernel shows broader, diffuse similarity structures, whereas the FGW kernel produces sharper and more localized patterns, consistent with its formulation that incorporates both feature and relational structure. These results imply that the FGW-based kernel provides a more expressive and efficient foundation for learning over CW complex distributions.

| Kernel | Test Loss (MLL) |
|---|---|
| $k_W$ (Eq. 18) | 10.169 |
| $k_{FGW}$ (Eq. 19) | 5.071 |

Table 1: Comparison of test loss (marginal log-likelihood, MLL) across different kernel choices.

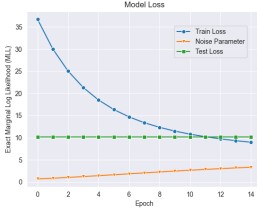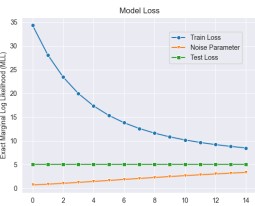

Figure 2: In the figure above we can see the training losses, noise parameter, and test loss change for each kernel. The left most graph corresponds to the exponential kernel from equation (18). The right most graph corresponds to the FGW kernel as in equation (19).

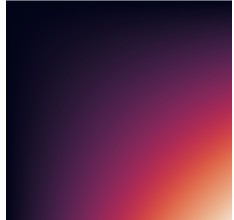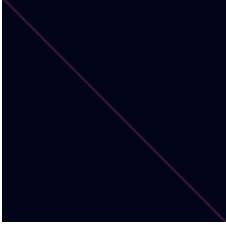

Figure 3: In the above figure we can see the two kernels visualized. On the left panel we see the exponential kernel from equation (18). The right-most panel is the FGW kernel as in equation (19).

# 6 Conclusion and Limitations

Thus we have derived an explicit expression for the Wasserstein distance between CW complex signal distributions. We introduced extensions of the Fused Gromov-Wasserstein distance to cell complexes. Then, we presented novel kernels over the space of probability measures on CW complexes. We have shown that when fitting a Gaussian Process leveraging the introduced kernels, one can approximate the optimal map which translates between cell complexes of the same dimension. We do not provide baseline comparisons with traditional graph or topological learning methods, as our focus is on defining and validating our transport-based framework. This is a key limitation of our work. We believe future work will explore applications of this framework to supervised learning tasks where comparative benchmarks are more appropriate. In short, we introduce a family of optimal transport kernels for cell complexes.

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

# A Appendix

## A.1 CW-Complex Definitions

Alain et al. (2023) define the first Gaussian process and two kernels on cell complexes. We provide their definitions and notation (Alain et al., 2023).

**Definition A.1.** Suppose $X$ is an $n$-dimensional complex. Then a $k$-chain $c_k$ for $0 \leq k \leq n$ is simply a sum over the cells: $c_k = \sum_{i=1}^{N_k} \eta_i e_i^k$, $\eta_i \in \mathbb{Z}$ The authors show this generalizes the notion of directed paths on a graph. The set of all $k$-chains on $X$ is denoted by $C_k(X)$ which has the algebraic structure of a free Abelian group with basis $\{e_i^k\}_{i=1}^{N_k}$.

**Definition A.2.** Given definition A.1, the boundary operator naturally follows as $\partial_k : C_k(X) \to C_{k-1}(X)$ which maps the boundary of a $k$-chain to a $k-1$-chain. This map is linear thereby leading to the equation: $\partial_k \left( \sum_{i=1}^{N_k} \eta_i e_i^k \right) = \sum_{i=1}^{N_k} \eta_i \partial(e_i^k)$.

The authors then define $k$-cochains and the coboundary operator. Each are dual notions of $k$-chains and the boundary operator respectively.

**Definition A.3.** Suppose $X$ is an $n$-dimensional complex. Then a $k$-cochain on $X$ is a linear map $f : C_k(X) \to \mathbb{R}$ where $0 \leq k \leq n$. $f \left( \sum_{i=1}^{N_k} \eta_i e_i^k \right) = \sum_{i=1}^{N_k} \eta_i f(e_i^k)$ where $f(e_i^k) \in \mathbb{R}$ is the value of $f$ at cell $e_i^k$. The space of $k$-cochains is defined as $C^k(X)$, and forms a real vector space with the dual basis $\{(e_i^k)^*\}_{i=1}^{N_k}$ such that $(e_i^k)^*(e_j^k) = \delta_{ij}$.

**Definition A.4.** Given definition A.3, the coboundary operator naturally follows as $d_k : C^k(X) \to C^{k+1}(X)$ which for $0 \leq k \leq n$ is defined as $d_k = f(\partial_{k+1}(c))$ for all $f \in C^k(X)$ and $c \in C_{k+1}(X)$. Note that for $k \in \{-1, n\}$ $d_k f \equiv 0$.

Building upon these definition, Alain et al. (2023) formally introduce a generalization of the Laplacian for graphs termed the Hodge Laplacian. The Hodge Laplacian is to CW-complexes, what the Laplacian is to graphs. In essence, the authors prove $W_0 = I \implies \Delta_0 = B_1 W_1 B_1^\top$ and $W_1 = I \implies \Delta_0 = B_1 B_1^\top$, thereby showing that the Hodge Laplacian is the standard graph Laplacian when $k = 0$ and one chooses identity weights.

**Definition A.5.** Let $X$ be a finite complex. Then we define a set of weights for every $k$. Namely let $\{w_i^k\}_{i=1}^{N_k}$ be a set of real valued weights. Then $\forall f, g \in C^k(X)$ one can write the weighted $L^2$ inner product as: $\langle f, g \rangle_{L^2(w^k)} := \sum_{i=1}^{N_k} w_i^k f(e_i^k) g(e_i^k)$. This inner product induces an adjoint of the coboundary operator $d_k^* : C^{k+1}(X) \to C^k(X)$. Namely $\langle d_k^* f, g \rangle = \langle f, d_k g \rangle$ for all $f \in C^{k+1}(X)$ and $g \in C^k(X)$.

**Definition A.6.** Using the previous definitions the Hodge Laplacian $\Delta_k : C^k(X) \to C^k(X)$ on the space of $k$-cochains is then defined to be $\Delta_k := d_{k-1} \circ d_{k-1}^* + d_k^* \circ d_k$. Which has matrix representation $\Delta_k := B_k^\top W_{k-1}^{-1} B_k W_k + W_k^{-1} B_{k+1} W_{k+1} B_{k+1}^\top$. Here, $W_k = \text{diag}(w_1^k, \ldots, w_{N_k}^k)$ is the diagonal matrix of cell weights and $B_k$ is the order $k$ incidence matrix whose $j$-th column corresponds to a vector representation of the cell boundary $\partial e_j^k$ viewed as a $k-1$ chain.

