# OpenReview forum: "Families of Optimal Transport Kernels for Cell Complexes"
_TMLR — Rejected by TMLR_

### Review · Reviewer_Hkan · 2025-05-27

**Summary Of Contributions:**

This work introduces an optimal transport framework between CW complexes, which are topological spaces that generalize simplicial complexes in algebraic topology. In particular:

1. This work proposes to place the smooth graph signal distributions, which are normal distributions based on the Hodge Laplacian matrices) on the two CW complexes and then calculate the Wasserstein distance between these distributions.
2. This work also proposes Fused Gromov-Wasserstein (FGW) distance to combine both the structural information from the complexes and the feature information from the cell's weights.

The authors then proceed to define new Gaussian process kernels based on these two Wasserstein distances. An experiment on simulated CW complexes with cell weights as features shows that FGW yields a better test performance, in terms of marginal log-likelihood, in this setting.

**Audience:**

Yes

**Broader Impact Concerns:**

No impact concerns.

**Claims And Evidence:**

No

**Requested Changes:**

Please clarify or make the following fixes:
- 1.1. What is $n$?
- 1.1 Is the $N$ in $\\{ e^j_i \\} \_{i=1}^N$ the same as $N$ in $\\{e^0_i \\}_{i=1}^N$ i.e. same number of cells?
- 1.1 What is a "skeleton"?
- 2.2 $L_{ijkl}$ can simply be replaced by $L(C_1,C_2)$
- 3.1 What's the definition of "topological dimension"?
- 3.4 "information of we" there seems to be a typo here.
- Equation (16) $ \Pi(\mu, \mu) $ should be  $\Pi(\mu, \nu)$
- Equation (22) $\theta$ appears twice.
- Equation (22) what are $\mu_i$'s and $\nu_i$'s ?
- Equation (22) I am not familiar with this definition of optimal transport (with $\nu_i$ conditional on $\mu_i$). Could the authors expand a bit more on how this definition is related to the well-known definition?

**Strengths And Weaknesses:**

## Strenghts

- The paper nicely review all the previous methods for optimal transport on graphs and structured data, then subsequently introduces how these methods can be used for CW complexes.

## Weaknesses

- I personally find the topic to be interesting. However, there are many parts in this paper that does not provide sufficient motivation for wider audience to care about CW complexes. Starting with the introduction, the authors should provide some real-world examples to motivate why we should care about cell complexes. The recommended writing style for the introduction, in no particular topic order, is as follows:
    - Whats and whys of CW complexes. How are they related to graphs and structured data?
    - Why optimal transport for CW complexes? What are applications?
    - The current advance in optimal transport for graphs and structured data, and how they can be used for CW complexes.
  In general, I recommend taking a look at Vayer et al. (2019) and Alain et al. (2024) for references on writing style.
- For a paper that focuses on algebraic topological data, there ought to be more visualizations of CW complexes and their transports. Without them, it is hard to visualize myself how these objects look like.
- Lacks of motivation to any objects defined in this paper. For example, why use the smooth graph signal distributions on CW complexes? What are their interpretation and motivation?
- I see that the Wasserstein and FGW distances proposed in this work are simple extensions of those from Petric Maretic et al. (2019) and Vayer et al. (2019), respectively, with the difference being the definition of the Hodge Laplacian matrix. Could the authors expand on whether the theoretical results in the aforementioned references hold under this setting as well?
- I think the Polish space formulation is not necessary and makes the paper overly complicated. In particular, as the optimal transports are performed on the space of normal distributions with the pseudo-inverse of Hodge Laplacian matrices, we are essentially working in a finite-dimensional Euclidean space. So there is no need for the notion of Polish space unless there are some theoretical results that require it.

  My suggestions would be to replace the Polish space-related content by illustrative exposition of CW complexes that allow general audience to grasp the concept, as I described above.
- Even though the authors repeatedly mention that they will not compare with Petric Maretic et al. (2019), Vayer et al. (2019) and Alan et al. (2024)'s methods since "their focus is on defining and validating their transport based framework". I personally think that this is not the right mindset when presenting a new framework. This work suggests that using full information of CW complexes should result in a better transport than the previously proposed methods under a right metric that is suitable for CW complexes. In particular, the standard Wasserstein distance between graphs, which does not use the feature information, should definitely perform worse than the proposed method here. I suggest that, at least, the authors compare their proposed distances with the original Wasserstein and FGW distances.
- In the experiment, only simulation is fine, but I would like to see at least one downstream task that can utilize the proposed method on real-world data.

All in all, I find the potential application of algebraic topology to real-world problems very interesting, but the writing and organization require a major overhaul to be more accessible to general audience.

---

> ### Author Response · Authors · 2025-06-18
> **Re: Reviewer Hkan**
>
> 1. **Motivation**
> - We fully agree with the reviewer’s suggestions. To address these points, we will:
> - rewrite the introduction to clearly explain what CW complexes are, how they generalize graphs and simplicial complexes, and why they are relevant in structured data analysis and ML.
> - provide intuitive explanations and illustrative examples, supported by figures, to help readers visualize CW complexes and their topological features.
> - discuss why optimal transport is a natural tool for comparing CW complexes, emphasizing its ability to incorporate both structural and feature information.
>
> 2. **Visualization**:
> - We agree and can add a figure depicting the optimal transport paradigm in the context of CW complexes
>
> 3. **Motivation**:
> - Why use the smooth graph signal distributions on CW complexes? What are their interpretation and motivation?
> - We use the Hodge Laplacian as it is the primary component that provides structural information about the complex. It's somewhat analogous to the laplacian on graphs which is used in graph optimal transport.
> - We can add more to the intro addressing interpretation and motivation.
>
> 4. **Theoretical Results**
> - Yes the same theoretical guarantees should hold. To be a bit more specific can the reviewer identify which theorems or lemmas we can prove still hold from Petric Maretic et al. (2019) and Vayer et al. (2019)? We can add a proof to either the appendix or main results depending on where the reviewer feels it would be most appropriate.
> - As for the key differences, In the case of graphs there is only one laplacian matrix used via the distance metric. In the case of cell complexes one doesn't have just one Hodge Laplacian matrix. There are $k$-many Hodge Laplacians for 1 cell complex of dimension $k$. Hence we need to incorporate a distance that works well/preserves info in much higher dimensional (non-Euclidean) space. The Alain paper expands on this pretty well [3]. One can reference Sections 4.2 and 4.3 of that paper for further explanation.
>
> 5. **Polish space formulation**
> - We acknowledge the concern but choose this formulation to align with foundational OT literature Cédric Villani [4]. Essentially, he formulates it from the standpoint of Polish spaces so we adopt that convention. The decision to present the work from this standpoint was to align with the literature.
>
> 6. **Experiment/Real-World**:
> - This is complex and nuanced. A few things should be highlighted here.
> - The proposed task: learning or comparing distributions on CW complexes is new, and as such, there are no existing real-world CW-complex datasets or baselines.
> - Essentially there is no existing gaussian process/ML model that maps from Cell complex $\to$ Cell complex.
> - There are no existing/provided datasets where one learns a mapping from Cell complex $\to$ Cell complex.
> - Therefore we went with synthetic data (1000 samples) to demonstrate that the proposed kernels can distinguish structure and features meaningfully.
> - However, CW complexes are becoming increasingly common learning representations in cheminformatics, bioinformatics, and even NLP. One can have a dataset of molecules and represent each molecule as a Cell complex. However going from Cell complex
>  Cell complex is analogous to learning how a molecule transforms in a molecular dynamics simulation/trajectory. That ground truth dataset doesn't exist at all.
> - The actual task/experiment conducted "learning the OT map for cell complexes" is novel. Essentially, we are the first to develop a model which can even complete such a task.
>
> References:
> 3. https://icml.cc/virtual/2024/poster/33677
> 4. https://www.math.ucla.edu/~wgangbo/Cedric-Villani.pdf

---

> ### Author Response · Authors · 2025-06-18
> **Re: Clarifications**
>
> 1.1. What is $n$?
> - This is the dimension of the complex.
> 1.1 Is the $N$ in $\{ e^j_i \} _{i=1}^N$ the same as $N$ in $\{e^0_i \}_{i=1}^N$ i.e. same number of cells?
> - number of cells yes but may be different for different dimensions maybe $N_j$ and $N_0$ is clearer.
> 1.1 What is a "skeleton"?
> - backbone of a cell complex (graph without loops)
> 2.2 $L_{ijkl}$ can simply be replaced by $L(C_1,C_2)$
> - Yes but that diverges from the convention in Vayer/Petric Maretic. I can state the equivalency and convention in the paper if its clearer.
> 3.1 What's the definition of "topological dimension"?
> - The largest natural number $n$ for which there are non-trivial $n$ cells. [https://ncatlab.org/nlab/show/dimension+of+a+cell+complex]
> 3.4 "information of we" there seems to be a typo here.
> - We can correct that. Thank you for pointing this out!
>
> Equation (16) $ \Pi(\mu, \mu) $ should be $\Pi(\mu, \nu)$
> - We can correct that. Thank you for pointing this out!
> Equation (22) $\theta$ appears twice.
> - We can correct that. Thank you for pointing this out!
>
>
> Equation (22) what are $\mu_i$'s and $\nu_i$'s ?
> Equation (22) I am not familiar with this definition of optimal transport (with $\nu_i$ conditional on $\mu_i$). Could the authors expand a bit more on how this definition is related to the well-known definition?
>
> - these two follow from section 3 of the paper: https://proceedings.neurips.cc/paper_files/paper/2016/file/2a27b8144ac02f67687f76782a3b5d8f-Paper.pdf
> - It's more in the spirit of Rasmussen & Williams (2006) aka GP training by maximizing expected marginal log likelihood
> - Also from OT-solver literature by Genevay et al. (2016) [linked above] -> the actual notation  $\mu_i$'s and $\nu_i$ is re-used from there.

---

> > ### Comment · Reviewer_Hkan · 2025-06-19
> > **Reply**
> >
> > Thanks for the response. I have some remaining concerns.
> >
> > > We use the Hodge Laplacian as it is the primary component that provides structural information about the complex.
> >
> > Thanks. What is the motivation behind using the normal distribution with the pseudo inverse of the Hodge Laplacian as the covariance matrix?
> >
> > > Can the reviewer identify which theorems or lemmas we can prove still hold from Petric Maretic et al. (2019) and Vayer et al. (2019)?
> >
> > Specifically, can you explain why Theorem 3.1 and 3.2 in Vayer et al. (2019) hold in the more general setup? I understand that you mention Theorem 3.1 in the paper, but I want to know why the proofs for those theorems hold for this paper as well.
> >
> > > We acknowledge the concern but choose this formulation to align with foundational OT literature Cédric Villani.
> >
> > Okay, can you explain how the Euclidean space Y obtained from the Whitney's theorem is related to the calculation done in the experiment? Since the OT distances defined in this paper solely depend on this embedding. Or can we somehow calculate the distances without knowing Y?
> >
> > An experiment using cheminformatics/bioinformatics would be nice. It would make this paper more convincing.

---

> > > ### Author Response · Authors · 2025-06-25
> > > **Re: Reply**
> > >
> > > 1. Motivation behind the normal with the pseudo inverse of hodge laplacian?
> > >
> > > Thats a phenomenal question! The paper that answers that is [1]. Essentially, in graphs the pseudo-inverse formulation means that the graph signal values vary slowly between strongly connected nodes [1]. That assumption is verified for most common graph and network datasets. It is further used in many graph inference algorithms implicitly representing a graph through its smooth signals [2].
> > >
> > > The high level answer is that we believe that its the best way to represent the local structure through the smooth signals. Essentially what worked for graphs is a reasonable assumption for cell complexes but it hasn't been verified in the same way for cell complex datasets because almost none of those exist. Verification would require another paper similar to [1].
> > >
> > > 2.  Theorem 3.1 and 3.2 in Vayer et al. (2019)
> > >
> > > Note: this is kind of hand-wavy / intuition not a formal claim in the paper
> > >
> > > (theorem 3.1): I believe this has to do with the interpolation property w.r.t $\alpha$ so essentially:
> > > (1) we mirror the expectation term as in Vayer
> > > (1.1) the feature component $(1-\alpha)M_{k,X_1,X_2}^{p}$ refers to the distance matrix between cell weights
> > > (1.2) the structure component $\alpha L(\Delta_{X_1}, \Delta_{X_2})^{p}$ is the similarity tensor between the structures
> > > since the actual functional form is an analogue:
> > > - as $\alpha \to 0$ the structure term vanishes therefore we have the wasserstein distance between the distributions of cell weights. This is the limit for the FGW distance.
> > > - as $\alpha \to 1$ the feature term vanishes. Therefore only the structure term, recovers the Gromov wasserstein distance between the structures.
> > >
> > > (theorem 3.2) Vayer for $q > 1$ the FGW distances defines a semi-metric and for $q = 1$ we get a metric under some isometry conditions. This holds in our case because we rely on our distances being well defined semi-metrics. The key change is in the domain of the distance function. There's a required relaxation though w.r.t triangle inequality that you have to prove.
> > >
> > > To actually prove these things would require more space and rigor/careful construction and a few supporting lemmas. The intuition is there though.
> > >
> > > 3. Experiment: how the Euclidean space Y obtained from the Whitney's theorem is related to the calculation done in the experiment? Since the OT distances defined in this paper solely depend on this embedding. Or can we somehow calculate the distances without knowing Y?
> > >
> > > We basically know from the whitney/menger argument that one **can** embed nicely into a euclidean space. In practice we actually do just use a simple embedding layer on the tensors to do this. we do need Y to calculate the distances your intuition is correct. We do specify that the learned optimal transport map is approximate for this among other reasons.
> > >
> > > [1] X. Dong, D. Thanou, P. Frossard, and P. Vandergheynst. Learning laplacian matrix in smooth
> > > graph signal representations. IEEE Transactions on Signal Processing, 64(23):6160–6173,
> > > 2016.
> > >
> > > [2] Petric Maretic, Hermina, et al. "GOT: an optimal transport framework for graph comparison." Advances in Neural Information Processing Systems 32 (2019).

---

> > > > ### Comment · Reviewer_Hkan · 2025-06-27
> > > > **Thanks**
> > > >
> > > > Thanks. I think that your answers to  Question 1 and 3 should be included in the paper. The proofs of the theorems in Question 2 would be nice, but they are not as important. For Question 3, you mentioned "a simple embedding layer"; I think this is an important step that is omitted from the paper. As a whole, I believe the paper would greatly benefit from a better exposition and transparency in its implementation.

---

### Review · Reviewer_HQfn · 2025-05-27

**Summary Of Contributions:**

The paper presents a way of defining optimal transport distances between two CW complexes. This generalizes the concept of OT on graphs and simplicial complexes. The authors derive closed form expressions for Wasserstein distances between signal distributions on CW complexes. They extend Fused Gromov-Wasserstein distance to incorporate both cell features and the structural information. They show synthetic experiments on random CW complexes and show that FGW-based kernel outperforms Wasserstein kernel.

**Audience:**

No

**Broader Impact Concerns:**

NA.

**Claims And Evidence:**

Yes

**Requested Changes:**

The paper would be stronger with more illustrative examples for the concepts being developed and discussed in sections 3 and 4.

Please refer to the weaknesses section.

**Strengths And Weaknesses:**

Strengths:

The paper has a strong theoretical component, where the authors establish the necessary results to define OT distances on CW complexes.

Further, the authors construct new kernel families (Wasserstein and Fused Gromov-Wasserstein) over CW complexes, which, they show, can be used for learning on cell complexes.

Weaknesses:

Since a graph is a simplicial complex and a simplicial complex is a CW complex, the authors can provide a comparison of their method with existing methods on graphs and simplicial complexes.

While I understand that the paper presents a novel theoretical way to compare two CW complexes by the medium of optimal transport map, I do not think that the experiment section is substantial to support the need for such a framework.

One of the other concerns I have is that the authors do not clearly pinpoint any specific problem that was limited by and could not be solved before which can now be solved using OT distances on CW complexes.

I am also not fully convinced that most of the TMLR audience would be well-versed with the concepts described in the paper. They need further explanation and illustrative examples to be completely understood.

---

> ### Author Response · Authors · 2025-06-18
> **Re: Reviewer HQfn**
>
> We thank the reviewer for the detailed summary and for recognizing the novelty of applying optimal transport to CW complexes. Below, we address your comments point by point.
>
> 1. **Baseline Comparison/Experiments**:
> - The proposed task: learning or comparing distributions on Cell complexes is new, and as such, there are no existing real-world CW-complex datasets or baselines.
> - Essentially there is no existing gaussian process/ML model  that maps from Cell complex $\to$ Cell complex.
> - There are no existing/provided datasets where one learns a mapping from Cell complex $\to$ Cell complex.
> - Therefore we went with synthetic data (1000 samples) to demonstrate that the proposed kernels can distinguish structure and features meaningfully.
> - However, Cell complexes are becoming increasingly common learning representations in cheminformatics, bioinformatics, and even NLP. One can have a dataset of molecules and represent each molecule as a Cell complex. However going from Cell complex $\to$ Cell complex is analogous to learning how a molecule transforms in a molecular dynamics simulation/trajectory. That ground truth dataset doesn't exist at all (you would have to derive it -> that data would need to be validated by scientists in MD).
> - We acknowledge that graphs and simplicial complexes are special cases of CW complexes. However, existing OT methods tailored for graphs or simplicial complexes do not extend straightforwardly to the general CW complex setting, especially considering the richer cell attachment structures and topological complexity. Our framework operates on the Hodge Laplacian of arbitrary Cell complexes, capturing higher-order topological features that prior methods cannot. To clarify this distinction, we will add a detailed discussion.
> - If one wanted to provide a direct comparison to any other standard/existing GP method for graphs/simplicial complexes it would require probably adding linear layers in post which is a different kind of model (from the GP) and leads to a loss of structural information [2]. You can think of the linear layers as almost like dimensionality reduction/ lossy compression. You would map from Cell complex $\to$ vector/scalar instead of  Cell complex $\to$ Cell complex. To make that final transform you add the linear layers or use a different method. You would have to add these layers because (to the best of my knowledge) every existing ML method for Cell complexes maps from a Cell complex $\to$ vector/scalar.
> - We will clarify this distinction and discuss potential baseline comparisons in the revision.
>
> 2. **Need/Specific Problem/Background**
> -  We agree that a more general explanation of the motivation would be a necessary step as well as adding more background on cell complexes. We plan to take the following concrete steps:
> - rewrite the introduction to mention motivating real-world examples particularly from chemistry and materials science. (Essentially these higher-order cell complex structures show up more naturally in this setting and there is a severe lack of learning methods (few choices) for cell complexes).
> - add intuitive background explanations for CW complexes, differences between simplical complexes and why they are meaningful in ML.
> - add in figures explaining what cell complexes are and how to construct them.
>
> References:
> 2. Maosheng Yang and Elvin Isufi. Hodge-aware learning on simplicial complexes, 2023. URL https://
> openreview.net/forum?id=QSJKrO1Qpy.

---

### Review · Reviewer_7cqH · 2025-06-08

**Summary Of Contributions:**

This work introduces a way to compare cell complexes using optimal transport (OT) methods, based on Hodge-Laplacian matrices and comparing Gaussian of the signals, in the same spirit as [1] on graphs. Moreover, this work also proposes to take into account features by generalizing the Fused Gromov-Wasserstein to this setting. Then, it also introduces Gaussian kernels using the two distances. Finally, it is proposed to find the OT map through Gaussian processes.

[1] Petric Maretic, H., El Gheche, M., Chierchia, G., & Frossard, P. (2019). GOT: an optimal transport framework for graph comparison. Advances in Neural Information Processing Systems, 32.

**Audience:**

Yes

**Claims And Evidence:**

Yes

**Requested Changes:**

There are few things that could improve this work. First, I would recommend to add a more general introduction. For now, the paper dives directly into the maths, and it is thus hard to follow, or to understand why we want to compare cell complexes...

Personally, I know OT but not much on topological machine learning. Thus, the very fast introduction of cell complexes is hard to understand. It would be better to add a little more background on it. Concerning the background on OT in Section 1.2, this part could also be greatly improve. It is for now only a list of definitions, which might not be clear to a reader not familiar with OT. I would recommend to add a descriptions of the concepts, which is done in part in the intro of Section 5 with the description of the Monge problem, but I believe it should be also in the introduction. Moreover, the citations for some concepts are a bit weird. Why do you cite (Genevay et al, 2016; Vayer et al, 2018) for couplings in Definition 1.1, and (Vayer et al, 2018) for the Monge problem in Definition 1.2?

In Section 3, it is not clear to me how the embedding theorem works. Is the full CW complex represented as a point in a Euclidean space? Or is it represented as a probability distribution over an Euclidean space? This could be made clearer e.g. in Definition 3.3 by writing $OT_c(X,Y)=...$ if the comparison is between CW complexes, or $OT_c(\mathbb{P},\mathbb{Q})$ with $\mathbb{P},\mathbb{Q}$ distributions over CW complexes if is defined between probability distributions of CW complexes.

The Hodge-Laplacian matrix is a very important object in this work, but only introduced in Appendix. I feel it would deserve to be in the background part of CW complexes.

In Definition 3.4 and 3.5, all the notations are reminded (Hodge-Laplacian, Moore-Penrose inverse, Polish embeddings...). If these notions are well introduced in a background part, these definitions could be shorter and focus to the main points, which I believe would be clearer.

In Definition 3.5, the FGW distance is introduced to compare CW complexes. Why don't you also consider the Gromov-Wasserstein distance, which would be more comparable to the distance of Definition 3.4, as it would not take into account the features.

In Section 4, there are no references for the claims that the kernels are not positive definite. Morerover, Section 4.1 and 4.2 are almost the same. I would recommend to merge them.

In the abstract, it is claimed that "we introduce novel kernels over the space of probability measures on CW complexes based on the dual formulation of optimal transport." But I don't think the dual is ever again mentioned. Also it is claimed that the kernels over CW complexes rely on regularized OT in Section 2.1 and at the end of Section 4. But it is not mentioned after? Moreover, regularized OT commonly refers to entropic regularized OT, which does not seem to be used here.

The approximation of the OT map by fitting a GP is very interesting in my opinion, but I don't really understand equation (22), and the explanations below. For instance, what is $P(\nu_i|f_\theta(\mu_i),\mu_i)$ referring to? Same question for $N(\nu_i|f_\theta(\mu_i),\sigma^2)$?

Concerning the experiments, it would have been better to at least use the kernels to solve some tasks such as classification or regression. And it would also be a nice addition to consider real data, as for now, the experiments seem to be restricted to synthetic data?


**Typos:**
- Equations 4, 5, 13, 14: lack square on $W_2$
- Page 5: "compare structural information of we"
- Equation (18): lack square on $W_p$?
- Equation (22): don't need the constraint $\theta\in\mathbb{R}^{n\times k}$ which is mentioned twice.

**Strengths And Weaknesses:**

This work proposes to use optimal transport to compare cell complexes, which is an interesting contribution. However, the paper is not well written nor very clear, and lacks of motivations or of motivating applications.

**Strengths:**
- Optimal transport is proposed to compare cell complexes, either using OT between Gaussian representing signal over a cell complex, or using a Fused Gromov-Wasserstein distance.
- The perspective of using Gaussian Processes to approximate the OT map seems interesting.

**Weaknesses:**
- There is a lack of motivation (no real introduction, no task for this problem...)
- The paper is not well organized, not well written, and not very clear.
- The experiments are not really convincing.

---

> ### Author Response · Authors · 2025-06-18
> **Re: Reviewer 7cqH**
>
> We thank the reviewer for the detailed summary and for recognizing the novelty of applying optimal transport to CW complexes, as well as the interest in the GP-based OT map approximation. Below, we address your comments point by point.
>
> 1. **Introduction:** We agree that a more general introduction would be a necessary step as well as adding more background on cell complexes. We plan to take the following concrete steps:
> - rewrite the introduction to mention motivating real-world examples particularly from chemistry and materials science. (Essentially these higher-order cell complex structures show up more naturally in this setting and there is a severe lack of learning methods (few choices) for cell complexes).
> - add intuitive background explanations for CW complexes, differences between simplical complexes and why they are meaningful in ML.
> - We will correct the citations in definitions 1.1 and 1.2 with clear framing.
>
> 2. **Embeddings**: We agree that more clarity could be added.
> - Each CW complex is represented by a Gaussian distribution over a signal space induced by the Hodge Laplacian.
> - We can make this clearer and explicitly write the distance defined between the measures induced not the points.
>
> 3. **Question**: Why not the Gromov-Wasserstein Distance?
> - We can define this and make this comparison. We are open to adding another distance. It would be more comparable to 3.4.
> - We include Fused because it naturally combines structure and features. Gromov-Wasserstein is mostly structure only which we handle via Wasserstein over the signals.
>
> 4. **Background/Merging**:
> - We can add the notation/concepts for (Hodge-Laplacian, Moore-Penrose inverse, Polish embeddings...)  to the background/intro section.
>
> 5. **Section 4**:
> - We can definitely merge the sections 4.1 and 4.2.
> - It's definitely the case that the distance won't result in a positive definite kernel however; it's necessary to use the trick. The standard Wasserstein exponential kernel isn't positive definite even for point clouds / euclidean space [1]. In our more-general framing this is definitely the case (encompasses the euclidean case).
>
>
> 6.**Abstract**:
> - You're correct... We think dual is a mistake/typo and that shouldn't be regularized OT bc the in the literature it refers to entropic.
> - We will make those corrections.
>
> 7. **the OT map**:
> - Thank you for saying it's interesting. In fact, learning the OT map via a GP for cell complexes is a novel task. We're the first to do it.
> - As for the notation:
> - you caught a typo the $N(\nu_i | f_{\theta}(\mu_i), \sigma^2)$  should actually be $\mathcal{N}(\nu_i | f_{\theta}(\mu_i), \sigma^2)$
> - The equation on the 10th line of our algorithm aka $J(\theta)$ is the expected marginal log likelihood (discrete) from bayesian statistics.
> - $P$ is just the probability function/measure defined on a set of events in a $\sigma$ algebra satisfying measure properties.
>
> 8. **Experiments**:
> - This is complex and nuanced. A few things should be highlighted here.
> - The proposed task: learning or comparing distributions on CW complexes is new, and as such, there are no existing real-world CW-complex datasets or baselines.
> - Essentially there is no existing gaussian process/ML model $f$ that maps from Cell complex $\to$ Cell complex.
> - There are no existing/provided datasets where one learns a mapping from Cell complex $\to$ Cell complex.
> - Therefore we went with synthetic data (1000 samples) to demonstrate that the proposed kernels can distinguish structure and features meaningfully.
> - However, CW complexes are becoming increasingly common learning representations in cheminformatics, bioinformatics, and even NLP. One can have a dataset of molecules and represent each molecule as a Cell complex. However going from Cell complex $\to$ Cell complex is analogous to learning how a molecule transforms in a molecular dynamics simulation/trajectory. That ground truth dataset doesn't exist at all.
>
> 9. **Typos**:
> - Thank you for the catches. We will fix all typos.
>
> References:
> 1. https://arxiv.org/abs/2002.01878

---

### Decision · Action_Editor_6YFs · 2025-07-21

**Recommendation:** Reject

**Additional Comments:**

If the authors want to resubmit the paper, I strongly encourage them to write the paper in a more didactic way, maybe starting from a practical ML problem that CW complexes can help solve. Adding more extensive experiments on real data demonstrating the merits of the proposed method compared to existing baselines would also greatly strengthen the paper.

**Audience:**

Yes

**Audience Explanation:**

The main idea of the paper is appealing, especially for researchers working at the intersection of ML and topological data analysis. However, the current presentation of the paper does not make it accessible enough for researchers who are not already familiar with the concepts of the paper. Similarly, the motivation behind this work and its usefulness to the ML community could be made clearer. The absence of real-world examples also diminishes its practical impact and appeal to ML researchers.

**Claims And Evidence:**

No

**Claims Explanation:**

This paper presents an interesting theoretical framework, introducing new methods for learning on CW complexes. However, the exposition lacks motivation and clarity, as noted by all reviewers. Several concepts that are critical to the paper, like CW complexes or Hodge Laplacian, are not introduced with enough clarity, especially in the context of a machine learning paper. For instance, the paper would benefit from illustrative examples around these concepts. The experimental section only showcases synthetic experiments and the methods presented in this work are not compared to existing baselines. There is no demonstration of the pratical usefulness of the present method. As such, the claims of this paper are not backed by clear evidence.

**Resubmission Of Major Revision:**

The authors may consider submitting a major revision at a later time.